# Changes in Clinical Practice in Adherence to the 2014 American Thyroid Association Guidelines on Thyroid Cancer: A Retrospective Study from a Tertiary Referral Center

**DOI:** 10.3390/jpm14070727

**Published:** 2024-07-05

**Authors:** Federico Cappellacci, Gian Luigi Canu, Eleonora Noli, Alessandro Argiolas, Giulia Peis, Maria Letizia Lai, Pietro Giorgio Calò, Fabio Medas

**Affiliations:** 1Department of Surgical Sciences, University of Cagliari, “Policlinico Universitario Duilio Casula”, 09042 Cagliari, Italy; gianl.canu@unica.it (G.L.C.); e.noli2@studenti.unica.it (E.N.); a.argiolas30@studenti.unica.it (A.A.); g.peis1@studenti.unica.it (G.P.); pgcalo@unica.it (P.G.C.); fabiomedas@unica.it (F.M.); 2Department of Cytomorphology, University of Cagliari, 09124 Cagliari, Italy; marialai@aoucagliari.it

**Keywords:** thyroid cancer, ATA, ATA guidelines, thyroid surgery

## Abstract

Thyroidectomy, a pivotal treatment for various thyroid disorders, has seen its indications evolve, particularly with the 2014 American Thyroid Association (ATA) Guidelines advocating for conservative surgical approaches like lobectomy. This retrospective study analyzes thyroidectomy practices at a high-volume center from January 2014 to December 2023, focusing on patients potentially eligible for lobectomy per ATA guidelines. The inclusion criteria were tumors < 4 cm, indeterminate thyroid nodules, or differentiated thyroid carcinoma with clinically uninvolved lymph nodes (cN0). This study analyzed the proportion of patients undergoing lobectomy versus total thyroidectomy (TT) and the oncological outcomes. Of 357 patients, 243 underwent TT and 114 underwent lobectomy. The prevalence of lobectomies rose markedly, comprising 73.9% of surgeries in 2023. TT patients were predominantly female (83.5%) and had higher rates of autoimmune thyroiditis (67.5%) and malignancy (89.7%). Lobectomy patients had larger nodules and more indeterminate cytology. Among 301 malignant cases, TT was associated with higher lymph node metastasis, but similar recurrence rates, compared to lobectomy. This study underscores a shift towards lobectomy, reflecting adherence to ATA guidelines and suggesting conservative surgery is feasible without compromising outcomes. Further research on long-term outcomes and refined patient selection criteria is needed to optimize surgical approaches.

## 1. Introduction

Thyroidectomy has long been a cornerstone in the management of various thyroid disorders [1,2]. Historically, the indications for this surgery have evolved in response to advancements in medical knowledge, surgical techniques, and diagnostic capabilities. A significant milestone in this evolution came with the release of the 2014 American Thyroid Association (ATA) Guidelines, which provided updated recommendations on the indications for thyroid surgery [3].

Particularly, these guidelines were driven by a growing body of evidence suggesting that more conservative surgical approaches could be appropriate for certain thyroid conditions, emphasizing a more nuanced approach to thyroid lobectomy and advocating for its use in specific clinical scenarios while considering factors such as tumor size, patient age, and the risk of malignancy. The same guidelines suggested lobectomies particularly in cases of cytologically indeterminate thyroid nodules or cN0 differentiated thyroid cancers, those smaller than 4 cm. Thus, the 2014 ATA Guidelines represent a shift towards tailored surgical strategies aimed at balancing the benefits of effective disease management with the potential risks associated with more extensive thyroid surgeries, such as total thyroidectomy [4,5,6]. In this article, we explore the changes in indications for thyroid lobectomy in our clinical practice, analyzing the activity and the surgical and oncological outcomes in our high-volume tertiary referral center for thyroid surgery.

## 2. Materials and Methods

This is a unicentric, retrospective study. We included patients who underwent total thyroidectomy or lobectomy at our center from January 2014 to December 2023. We included in our study patients who could potentially be treated with lobectomy according to ATA 2014 Guidelines; thus, inclusion criteria included a tumor size smaller than 4 cm (T1-T2), a preoperative diagnosis of indeterminate thyroid nodule or differentiated thyroid carcinoma, and clinically uninvolved lymph nodes (cN0). Exclusion criteria included patients with suspicious or preoperatively confirmed lymph node metastasis (LNM), medullary thyroid carcinoma, and patients who underwent a two-stage thyroidectomy due to a loss of signal at the intraoperative neuromonitoring (IONM) of the recurrent laryngeal nerve (RLN).

The objectives of our study were as follows:-To evaluate the impact in clinical practice of the ATA 2014 Guidelines; particularly, we evaluated the proportion of patients eligible for lobectomy according to the guidelines that actually underwent lobectomy versus total thyroidectomy during the study’s timeframe.-To evaluate the oncological outcome of lobectomy in terms of disease-free survival in patients who underwent lobectomy versus patients who underwent the complete removal of the thyroid gland.

In the preoperative evaluation of our patients, we included clinical history, physical examination, and blood tests to assess thyroid function and autoimmune thyroiditis. Fine-needle aspiration cytology (FNAC) was performed in all patients, and the results were classified according to the Consensus Statement of AIT (Italian Thyroid Association), AME (Medical Endocrinologist Association), SIE (Italian Endocrinology Association), and SIAPEC-IAP (Italian Society of Pathological Anatomy) for the Classification and Reporting of Thyroid Cytology [7]. Autoimmune thyroiditis was defined based on the presence of anti-thyroglobulin antibodies (Tg-Ab) or anti-thyroid peroxidase antibodies (TPO-Ab) and on the basis of histopathological examination.

High-resolution US of the neck was always performed before surgery by an experienced surgeon, with careful evaluation of the central and lateral compartments. In cases of advanced diseases, or for enlarged retrosternal goiter, a contrast-enhanced CT (Computed Tomography) scan was used to better evaluate the correct surgical option. Preoperative laryngoscopy was routinely performed to assess vocal fold mobility.

The surgical specimen was fixed with formaldehyde; sections were stained with hematoxylin–eosin (H&E) and analyzed by a dedicated pathologist. Immunohistochemical analysis was performed with the streptavidin–biotin technique on paraffin sections using anti-pan-cytokeratin antibody. Extrathyroidal extension was defined as the presence of a gross infiltration of perithyroidal tissues found in pathological examination. Vascular invasion was defined as the invasion of vessels in the tumor capsule or beyond it, with intravascular tumor cells attached to the vessel wall.

All patients were referred to an endocrinologist for postoperative management. Disease-free status was defined as showing no evidence of disease (no clinical evidence of tumor and no imaging evidence of disease via neck US; in addition, in patients that underwent total thyroidectomy, this was described as no imaging evidence of disease via RAI imaging and low serum Tg levels during TSH suppression [Tg < 0.2 ng/mL] or after stimulation [Tg < 1 ng/mL] in the absence of interfering antibodies). Disease-free survival was defined as the time elapsed from surgery to the detection of recurrent disease.

For the purposes of this study, we first compared patients who underwent total thyroidectomy versus those who underwent lobectomy. Subsequently, we performed a sub-analysis only including patients with histopathological diagnosis of malign thyroid tumors, comparing patients who had undergone complete thyroidectomy (i.e., total thyroidectomy or lobectomy and subsequent complete thyroidectomy) and patients who had undergone partial thyroidectomy (i.e., lobectomy).

The statistical analysis was performed with MedCalc^®^ vers. 22.017. The Chi-squared test was used for categorical variables. All continuous variables were examined and found to follow a non-normal distribution; thus, the Mann–Whitney test was used for continuous variables. The log-rank test was used to estimate the differences in Kaplan–Meier curves for independent risk factors. The results were considered statistically significant in the case of a *p*-value < 0.05. Continuous variables were expressed as median values and interquartile ranges (IQR).

## 3. Results

In the study period, 2923 patients were submitted to surgery for thyroid diseases in our unit. Of those, we included in our study 357 patients (12.2%) who underwent TT (n = 243; 68.1%) or lobectomy (n = 114; 31.9%) as their primary surgery. As reported in Table 1, and illustrated in Figure 1, the proportion of patients who underwent lobectomy increased progressively during the period of this study. The first lobectomies were performed in 2018. Over the next three years, the number of lobectomies and TTs performed were very close, whereas in 2022, we assisted with a greater number of lobectomies than TTs. In 2023, the last year of our study, lobectomies represented 73.9% of the total operations (34 out of 46 operations).

Table 2 provides a detailed comparison between patients who underwent TT and those who underwent lobectomy.

The median age of patients undergoing both TT and lobectomy was 49 years, with the interquartile ranges (IQR) being 37–59 for TT and 35–61 for lobectomy. The *p*-value of 0.96 indicates no statistically significant difference in age distribution between the two groups. Conversely, a significant difference was observed in the sex distribution between the groups: among the TT patients, 83.5% were female and 16.5% were male. In contrast, 70.2% of lobectomy patients were female and 29.8% were male, with a *p*-value of 0.0037. Autoimmune thyroiditis was significantly more prevalent in the TT group (67.5%) compared to the lobectomy group (43.0%), (*p* < 0.001). In addition, patients undergoing lobectomy tended to have fewer nodules; specifically, 62.3% of lobectomy patients had a single nodule, compared to 41.6% in the TT group. Conversely, having multiple nodules (2–4 or more than 5) was more common in the TT group (*p* = 0.0013). The median nodule size, as measured using ultrasound (US), was larger in the lobectomy group (18.2 mm) compared to the TT group (14 mm), with a *p*-value of 0.0145, indicating statistical significance. For several ultrasound nodule characteristics, there were no significant differences between the groups. Hypoechogenicity, irregular shape, and calcification rates were similar. However, smooth limits were significantly more common in the lobectomy group (34.2% vs. 13.2%; *p*-value < 0.001), and CDIII vascularization was more frequent in the TT group (46.9% vs. 34.2%; *p*-value 0.0237). The cytological findings showed significant differences: the TT group had higher percentages of Tir4 (41.6% vs. 18.4%) and Tir5 (29.2% vs. 7.0%) cytology results, while the lobectomy group had a much higher percentage of indeterminate thyroid nodules (74.6% vs. 29.2%); all these differences were statistically significant with *p*-values < 0.001. The median operative time was significantly longer for TT (80 min) compared to lobectomy (55 min), with a *p*-value < 0.001. Intraoperative neuromonitoring (IONM) use was similar between the two groups (90.1% for TT and 88.6% for lobectomy), with no significant difference.

Upon pathological examination, the TT group had a higher occurrence of malignancy (89.7%) compared to the lobectomy group (72.8%), and conversely, benign disease was more common in the lobectomy group (26% vs. 10.3%). The *p*-value for these differences was <0.001. The thyroid weight was significantly greater in the TT group (median 17 g) compared to the lobectomy group (median 9 g), with a *p*-value < 0.001.

Among the 357 patients included in our study, 301 with a pathological diagnosis of malignancy were included in a sub-analysis reported in Table 3, which provides a comparison between patients who underwent the complete removal of the thyroid gland (including TT as first surgery, or lobectomy followed by complete thyroidectomy) and those who underwent only partial thyroidectomy (i.e., lobectomy), focusing on demographic, clinical, and pathological characteristics, as well as surgical outcomes.

The median age of patients in both the total thyroidectomy and lobectomy groups was quite similar. We studied 239 patients (79.4%) in the complete thyroidectomy group (the median age was 47 years, with an interquartile range (IQR) of 35–58 years), while in the partial thyroidectomy group, we studied 62 patients (20.59%) and we observed that the median age was 49 years, with an IQR of 36–60 years. The *p*-value of 0.75 suggested no significant difference in age between the two groups. There was a significant difference in the sex distribution between the two groups. In the complete thyroidectomy group, 82.8% of the patients were female, compared to 70.9% in the other group. This difference, highlighted by a *p*-value of 0.0358, indicated that females were more likely to undergo complete thyroidectomy than partial thyroidectomy. Autoimmune thyroiditis was more prevalent in the complete thyroidectomy group, affecting 64.0% of the patients, compared to 50.0% in the other group. This difference was statistically significant, with a *p*-value of 0.0436. The median nodule size was 11 mm in both groups, with slight variations in the IQR (7–17 mm for complete thyroidectomy and 8–18 mm for partial thyroidectomy). The *p*-value of 0.97 indicates no significant difference in nodule size between the two surgical approaches.

The distribution of thyroid cancer subtypes showed some variability between the groups. Classic PTC was slightly more common in the complete thyroidectomy group (47.7%) compared to the other group (41.9%). The follicular variant of PTC was more prevalent in the partial thyroidectomy group (19.4%) than in the complete thyroidectomy group (13.8%). The tall cell variant of PTC was more frequent in the complete thyroidectomy group (20.5%) compared to the other group (9.7%). Follicular thyroid carcinoma was more common in the partial thyroidectomy group (20.9%) than in the complete thyroidectomy group (11.7%). The *p*-value of 0.0530 indicated a marginally significant difference in the distribution of histotypes between the two groups, suggesting potential variability in the underlying pathology might influence the choice of surgical procedure.

An extrathyroidal invasion was observed in 13.8% of the complete thyroidectomy group’s patients and 9.7% of the partial thyroidectomy group’s patients, with a *p*-value of 0.39, indicating no significant difference. Similarly, angioinvasion rates were comparable between the groups (5.4% for complete thyroidectomy and 4.8% for partial thyroidectomy), with a *p*-value of 0.85, suggesting no significant difference. Actually, lymph node metastasis was significantly more common in the complete thyroidectomy group (17.1%) compared to the partial thyroidectomy group (0%), with a *p*-value of less than 0.001. Recurrence rates were 2.5% in the complete thyroidectomy group and 0% in the other group. The *p*-value of 0.35 indicates no significant difference in recurrence rates between the two surgical approaches.

The log-rank test on the Kaplan–Meier curves (Figure 2), representing patients who underwent partial thyroidectomy and complete thyroidectomy, did not demonstrate a significant difference between the two groups (*p* = 0.17).

## 4. Discussion

This study provides a comprehensive analysis of the trends, demographic characteristics, clinical features, and outcomes associated with TT and lobectomy as primary surgical interventions for thyroid disease in our center. The findings reveal significant shifts in surgical practices over time and underscore the importance of tailored surgical approaches based on patient-specific factors and disease characteristics [8,9].

Particularly, our data illustrate a marked increase in the proportion of lobectomies over the study period, culminating in lobectomies accounting for 73.9% of surgeries in 2023. This shift reflects a progressive adherence of our center to the ATA guidelines and, consequently, a growing preference for more conservative surgical options when appropriate. This trend aligns with broader shifts in endocrine surgery, emphasizing the importance of preserving thyroid function when possible and minimizing surgical morbidity [6,8,10].

Our observation of the increasing preference for lobectomy over TT aligns with findings from several other studies. A study by Sosa et al. [11], including over 5000 patients, reported a similar trend towards increased lobectomy rates, attributing this shift to improved preoperative diagnostic techniques and a growing emphasis on less invasive surgical options for managing thyroid nodules and early-stage thyroid cancer. Other studies reported similar results [12,13,14,15,16].

It is important to highlight that our study period includes the years of the COVID-19 pandemic. The pandemic had a notable impact on the enrollment of cases for thyroid surgery in our unit, as well as many others in Italy and worldwide. We observed a clear reduction in surgical activity during the COVID-19 period; we preferred to give priority to the patients affected by thyroid cancer and those with advanced diseases. Overall, oncological activity was maintained [17].

Regarding demographic and clinical features, the median age of patients undergoing both TT and lobectomy was similar, indicating that age alone does not appear to be a primary factor in determining the type of surgery. However, a notable difference in sex distribution was observed: females were significantly more likely to undergo TT compared to males, which could be influenced by the higher incidence of thyroid disease in women and possibly differing clinical presentations between the sexes.

Autoimmune thyroiditis was significantly more prevalent in the TT group, suggesting that more extensive disease or associated autoimmune conditions may necessitate a more aggressive surgical approach. In contrast, patients undergoing lobectomy were more likely to present with fewer nodules, which may be more amenable to partial resection [18,19].

The demographic characteristics observed in our study, such as the higher prevalence of TT in females, are consistent with findings reported by Cinar et al. [20]. Their study also found that females are more likely to undergo TT, which may be related to the higher incidence of thyroid disorders in women.

The ultrasound characteristics revealed significant differences between the two groups. The lobectomy group had a higher proportion of patients with a single nodule and larger median nodule sizes, whereas the TT group had more patients with multiple nodules. Notably, smooth nodule margins were more common in the lobectomy group, while CDIII vascularization was more frequent in the TT group, suggesting that certain sonographic features may influence surgical decision-making.

The cytological findings further differentiated the groups. The TT group had higher rates of Tir4 and Tir5 cytology results, indicative of more suspicious or confirmed malignancies, which likely prompted more extensive surgery. In contrast, the lobectomy group had a higher incidence of indeterminate cytology, suggesting that in cases of diagnostic uncertainty, a less invasive approach is initially preferred [8,21].

Additionally, as previously highlighted, we observed that the median nodule size measured using ultrasound was significantly larger in the lobectomy group, which may initially seem unusual and contradictory. However, this can be attributed to the larger nodule size found in patients with indeterminate thyroid nodules, while patients with cytology results of Tir4 and Tir5 had smaller nodules [18]. Specifically, the median nodule size in patients with indeterminate thyroid nodules was 20 mm, compared to 12 mm in patients with Tir4 and Tir5.

The differences in ultrasound and cytological findings between the TT and lobectomy groups in our study are corroborated by research conducted by Filetti et al. (2018) [22]. Their study emphasized that certain sonographic features, such as larger nodule size and specific cytological results (e.g., Tir4 and Tir5), are predictive of malignancy and more aggressive disease, thus influencing the choice of TT over lobectomy. This aligns with our findings, which show that TT patients had higher rates of suspicious cytology and larger nodule sizes.

The pathological examination confirmed that the TT group had a higher prevalence of malignancy compared to the lobectomy group. This finding is consistent with the more suspicious cytological findings observed in TT patients. Conversely, benign conditions were more common in the lobectomy group, supporting the notion that lobectomy is often sufficient for less aggressive or uncertain thyroid lesions [21,23].

In our sub-analysis focusing on patients with malignancy, the distribution of thyroid cancer subtypes and other pathological features were examined. The complete thyroidectomy group had a higher incidence of lymph node metastasis; however, recurrence rates were low and did not significantly differ between the two groups, indicating that lobectomy might still be an adequate option for select patients with malignancy, particularly those with low-risk features.

In the sub-analysis of malignant cases, our observation that complete thyroidectomy was associated with higher lymph node metastasis rates compared to lobectomy is supported by a study conducted by Raffaelli et al. [24]. Their research demonstrated that more extensive surgical intervention is required in cases with lymph node involvement to achieve optimal oncological control.

There is an ever-increasing interest in the use of artificial intelligence in medicine. This could be a potential tool in thyroid surgery, allowing for better decision-making on the best surgical treatment for these patients. This is widely discussed in an interesting work by Taciuc et al. published in 2024 [25].

This study has some limitations. First, this is a single-center, retrospective study. The study was performed in an endemic iodine-deficient region, with a high incidence of autoimmune thyroiditis; therefore, the generalization of our results to other populations should be made carefully. Finally, it is crucial to emphasize that the incidence of recurrence observed in lobectomy cases has likely been underestimated. For patients who have undergone thyroidectomy, recurrence can be detected through an increase in Tg levels or changes in thyroid scintigraphy following RAI, in addition to neck ultrasound. However, for those who have had a lobectomy, recurrence can only be identified using neck ultrasound.

## 5. Conclusions

The findings of this study highlight the dynamic nature of thyroid surgery practices and the need for individualized patient management. The increasing preference for lobectomy suggests a shift towards more conservative management when appropriate, potentially reducing surgical morbidity without compromising oncological outcomes. Further research should focus on the long-term outcomes of lobectomy versus TT, particularly in terms of recurrence, survival, and quality of life. Additionally, prospective studies could help refine criteria for selecting the optimal surgical approach based on patient and disease characteristics. Additionally, further studies are needed to refine patient selection criteria for TT versus lobectomy, taking into account emerging molecular and genetic markers that can better stratify risk and guide surgical decision-making.

## Figures and Tables

**Figure 1 jpm-14-00727-f001:**
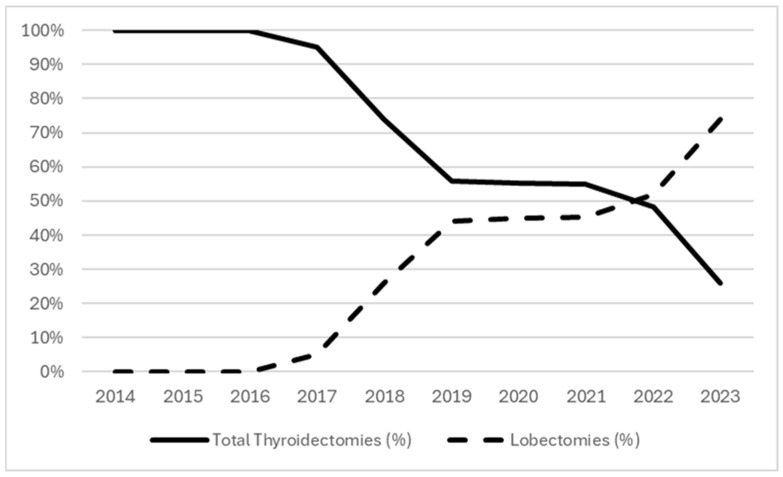
Trends in operations during the study period.

**Figure 2 jpm-14-00727-f002:**
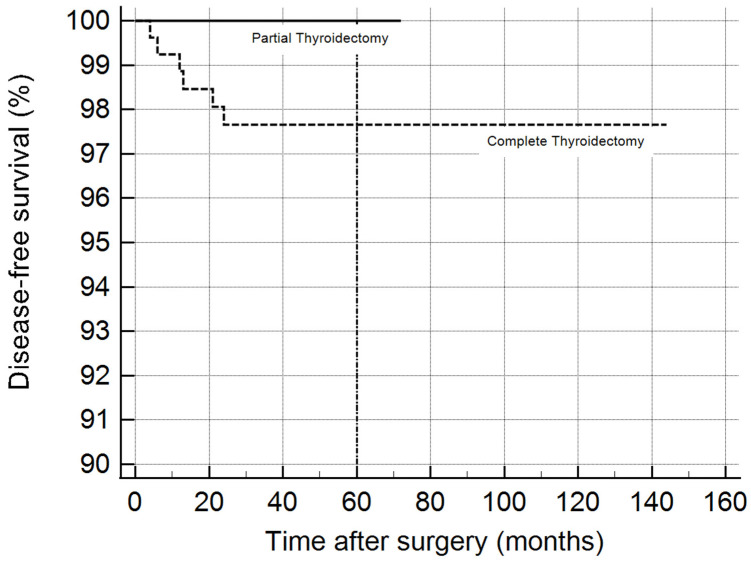
Kaplan–Meier curves estimating disease-free survival in patients who underwent partial thyroidectomy and complete thyroidectomy.

**Table 1 jpm-14-00727-t001:** Total thyroidectomies and lobectomies performed during the study period.

Year	Operations	Total Thyroidectomies	Lobectomies
2014	44	44 (100%)	0
2015	33	33 (100%)	0
2016	37	37 (100%)	0
2017	20	19 (95%)	1 (5%)
2018	27	20 (74.1%)	7 (25.9%)
2019	34	19 (55.9%)	15 (44.1%)
2020	29	16 (55.2%)	13 (44.8%)
2021	31	17 (54.8%)	14 (45.2%)
2022	54	26 (48.1%)	28 (51.9%)
2023	46	12 (26.1%)	34 (73.9%)

**Table 2 jpm-14-00727-t002:** Univariate analysis between total thyroidectomies (TT) and lobectomies.

	TT (n = 243)	Lobectomy (n = 114)	*p*
Age (year) median-IQR	49 (37–59)	49 (35–61)	0.96
Sex			
● Female	203 (83.5%)	80 (70.2%)	0.0037
● Male	40 (16.5%)	34 (29.8%)	
Autoimmune thyroiditis	164 (67.5%)	49 (43.0%)	<0.001
Number of nodules			
● Single	103 (42.4%)	71 (62.3%)	0.0013
● 2–4	103 (42.4%)	34 (29.8%)	
● >5	37 (15.2%)	9 (7.9%)	
Nodule size at US (mm)	14 (10-22)	18.2 (11–24)	0.0145
Main nodule features			
● Hypoechogenic	96 (39.5%)	46 (40.4%)	0.88
● Irregular shape	18 (7.4%)	11 (9.6%)	0.47
● Calcification	78 (32.1%)	39 (34.2%)	0.69
● Smooth limits	32 (13.2%)	39 (34.2%)	<0.001
● CDIII vascularization	114 (46.9%)	39 (34.2%)	0.0237
Cytology			
● Tir3-Tir3aTir3b	71 (29.2%)	85 (74.6%)	<0.001
● Tir4	101 (41.6%)	21 (18.4%)	<0.001
● Tir5	71 (29.2%)	8 (7.0%)	<0.001
Operative time (min)	80 (70–100)	55 (45–65)	<0.001
IONM use	219 (90.1%)	101 (88.6%)	0.66
Pathology			
Benign disease	25 (10.3%)	31 (26%)	<0.001
Malignancy	218 (89.7%)	83 (72.8%)	
Thyroid weight	17 (13–22)	9 (7–13)	<0.001

**Table 3 jpm-14-00727-t003:** Univariate analysis between complete thyroidectomies and lobectomies in patients with a diagnosis of thyroid carcinoma.

	Complete Thyroidectomies (n = 239)	Partial Thyroidectomies (n = 62)	*p*
Age	47 (35–58)	49 (36–60)	0.75
Sex			
● Female	198 (82.8%)	44 (70.9%)	0.0358
● Male	41 (17.2)	18 (29.1%)	
Autoimmune thyroiditis	153 (64.0%)	31 (50.0%)	0.0436
Nodule size (mm)	11 (7–17)	11 (8–18)	0.97
Histotype			0.0530
PTC classic	114 (47.7%)	26 (41.9%)	
PTC follicular variant	33 (13.8%)	12 (19.4%)	
PTC tall cell	49 (20.5%)	6 (9.7%)	
FTC	28 (11.7%)	13 (20.9%)	
Extrathyroidal invasion (ETE)	33 (13.8%)	6 (9.7%)	0.39
Angioinvasion	13 (5.4%)	3 (4.8%)	0.85
Lymph node metastasis	41 (17.1%)	0 (0%)	<0.001
Recurrence	6 (2.5%)	0 (0%)	0.35

## Data Availability

The data presented in this study are available upon request from the corresponding author. (The data are not publicly available due to privacy.)

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
