# Peer review of "Changes in Clinical Practice in Adherence to the 2014 American Thyroid Association Guidelines on Thyroid Cancer: A Retrospective Study from a Tertiary Referral Center"

_jpm, 2024, doi:10.3390/jpm14070727_

Round 1

Reviewer 1 Report

Comments and Suggestions for Authors

Dear Authors,

I read with great interest your article about current indications for thyroidectomy.

However, some aspects would need your attention.

There are many abbreviations in the text, please insert a list at the end of the study.

Your study is very complex on a retrospective 10 years, this period included also the COVID-19 pandemic, how did this affect the enrollment of cases in those 3 years?

It is good that you used surgeon-performed sonography for the management of the cases, but you should also mention other imaging modalities used in your protocol such as CT/MRI with contrast media in cases with advanced staging.

In the Discussion section, you need to mention the increased use of AI for personalized medicine. For example, AI-generated software could help the surgeon in deciding between the 2 treatment modalities in the future. Please reference this to the article by Taciuc IA, Dumitru M, Vrinceanu D, Gherghe M, Manole F, Marinescu A, Serboiu C, Neagos A, Costache A. Applications and challenges of neural networks in otolaryngology (Review). Biomed Rep. 2024 Apr 19;20(6):92. doi: 10.3892/br.2024.1781. PMID: 38765859; PMCID: PMC11099604.

Was there any case that required a conversion from lobectomy to thyroidectomy during surgery?

Please format the references according to MDPI format.

Looking forward to receiving the improved version of your manuscript.

Author Response

Dear reviewer, thank you very much for your effort in evaluating our work. All the suggestions you have made are uncontestable. Herein we include a point-by-point analysis of your observations:

This is the Abbreviation list that is now included in our work:

ATA: American Thyroid Association

TT: Total Thyroidectomy

LNM: Lymph Node Metastasis

RLN: Recurrent Laryngeal Nerve

FNAC: Fine-needle Aspiration Cytology

AIT: Italian Thyroid Association

AME: Medical Endocrinologist Association

SIE: Italian Endocrinology Association

SIAPEC-IAP: Italian Society of Pathological Anatomy for the Classification and Reporting of Thyroid Cytology

Tb_Ab: Anti-thyroglobulin Antibodies

TPO-Ab: Anti-thyroid Peroxidase Antibodies

US: Ultrasound

CT: Computed tomography

H&E: Hematoxylin-Eosin

RAI: Radioactive iodine

Tg: Thyroglobulin

IQR: Interquartile Range

IONM: Intra-Operative Neuromonitoring

PTC: Papillary Thryoid Cancer

FTC: Follicular Thryoid Cancer

ETE: Extrathyroidal invasion

The COVID-19 pandemic had a notable impact on the enrollment of cases for thyroid surgery in our unit. There is now a brief paragraph explaining how our numbers were affected by the COVID-19 pandemic

In cases of advanced diseases, or for enlarged retrosternal goiter, a contrast-enhanced CT scan was used to better evaluate the correct surgical option. This is now explained in our methods.

AI represents an interesting tool, as you correctly pointed out. We had inserted a paragraph regarding this topic in our discussion.

No intraoperative conversion from lobectomy to total thyroidectomy was necessary in our series.

References are now formatted according to the MDPI indications, we used the Zotero .csi available at the mdpi website.

Thank you again for the time dedicated evaluating our work

Reviewer 2 Report

Comments and Suggestions for Authors

I appreciate the chance to read and review the manuscript titled "Changes in clinical practice in adherence to the 2014 American Thyroid Association Guidelines on Thyroid Cancer: a retrospective study from a tertiary referral center" by Cappellacci et al. I thoroughly enjoyed reading this publication.

The abstract and introduction are uncontestable; they efficiently introduce the reader to the issues described in the manuscript.

The methodology is unquestionable: the project was meticulously planned, and proper statistical procedures were employed to analyze the gathered data. Data was acquired from a substantial number of patients, specifically 357 patients, of whom 68.1% underwent total thyroidectomy and 31.9% underwent lobectomy.

Regarding the described results (most of which cannot be criticized), I have a few comments that I would like to ask the authors to consider:

1.) Line 112 contains a typographical error, incorrectly using the value of 13.9% instead of the correct number of 31.9%. Please rectify this problem, as it is a benign but noticeable mistake because the values do not add up to 100%.

2.) Table 1 and Figure 1 present de facto identical data - this may not be strictly necessary, but I would consider including the table in the content of Figure 1, or possibly modifying Figure 1 so that it also presents the numerical data presented in the table.

3.) Please verify that all the data in Table 2 is correct. Specifically, please verify the "number of nodules" element for the group of patients who underwent total thyroidectomy; 2 individuals are either missing or did not qualify for either category (?). Taking this into account seems necessary because it may influence the results of the statistical analysis (although this seems unlikely, given the low p value in this particular data range).

The authors conducted the discussion correctly and included the proper references. I suggest incorporating information that clarifies the percentage of surgical procedures used, and why the authors' unit performed its first lobectomy in 2017.

The authors approached the limitations of their study appropriately. However, I recommend that they include details about their future research plans in this field. This would help minimize the potential for inference errors by readers that may arise from these limitations.

Taking into account the above comments regarding results and discussion / conclusions may, in my opinion, contribute to improving the quality of this already good manuscript.

Once the authors answer my comments, I am prepared to review the work again.

I wish the authors continued success in their scientific endeavors.

Author Response

Thank you very much for your comments regarding our work.

We have improved our manuscript based on your suggestion.

In details:

  1. Thank you for highlighting that, It was a typographical error and we have corrected it
  2. You are correct, the table and the figure represent the same data, we preferred to keep them separate because we believe it is clearer
  3. There was a typing error in Table 2, we have corrected in the improved version of the manuscript

In the study period, 2923 patients were submitted to surgery for thyroid diseases in our unit. 12.2% of these met our inclusion criteria and were included in the study. This is now specified in the results. As regards the first lobectomy, this could be explained by the strict inclusion criteria used for the study.

Regarding your last point, in our conclusions, we specify that further studies, ideally on genetic and molecular markers, could play a role in decision-making for these patients.

Thank you again for evaluating our work and for your suggestions, which have certainly improved our manuscript.

Round 2

Reviewer 2 Report

Comments and Suggestions for Authors

Thank you for the opportunity to re-evaluate the manuscript.

The authors responded positively to most of the recommended changes (with the exception of combining the presentation of results from the Table 1 and Figure 1, which does not reduce the quality of the manuscript but slightly limits the quality of results presentation, in my opinion).

I recommend accepting the article in its current form.